# Psychosocial Working Conditions in School and Life Satisfaction among Adolescents in Sweden: A Cross-Sectional Study

**DOI:** 10.3390/ijerph18105337

**Published:** 2021-05-17

**Authors:** Joakim Wahlström, Sara Brolin Låftman, Bitte Modin, Petra Löfstedt

**Affiliations:** 1Centre for Health Equity Studies (CHESS), Department of Public Health Sciences, Stockholm University, SE-10691 Stockholm, Sweden; sara.brolin.laftman@su.se (S.B.L.); bitte.modin@su.se (B.M.); 2Department of Public Health and Community Medicine, Sahlgrenska Academy, University of Gothenburg, Box 100, 40530 Gothenburg, Sweden; petra.lofstedt@folkhalsomyndigheten.se

**Keywords:** school demands, teacher support, classmate support, life satisfaction, positive health, Sweden

## Abstract

Poor psychosocial working conditions in school have consistently been shown to be associated with adverse health among adolescents. However, the relationships between school demands, teacher support, and classmate support and positive aspects of health have not been explored to the same extent. The aim of this study was to examine differences in psychosocial working conditions in school and in life satisfaction by gender and by grade, and to investigate the association between psychosocial working conditions in school and life satisfaction among boys and girls, and among students in different grades. Data from the Swedish Health Behavior in School-Aged Children (HBSC) study of 2017/18 were used, consisting of 3614 students in Grades 5, 7, and 9 (~11, 13, and 15 years). Psychosocial working conditions in school were captured by indices of perceived school demands, teacher support, and classmate support. Life satisfaction was measured by the 11-step Cantril’s ladder (using cutoffs at >5 and >8, respectively). Whereas girls reported higher school demands than boys, higher levels of teacher and classmate support were reported by boys. Students in lower grades reported lower school demands but higher levels of teacher and classmate support compared with students in higher grades. Boys and students in lower grades were more likely to report high life satisfaction compared with girls and students in higher grades. Results from binary logistic regression analyzes showed that school demands were inversely associated with life satisfaction, and that higher levels of teacher support and classmate support were associated with high life satisfaction. These results were found for both boys and girls, and for students in all grades. The findings indicate that schools have the potential to promote positive health among students.

## 1. Introduction

The youth years are marked by rapid and significant changes in life circumstances, including decreased dependency on parents in favor of relationships with peers as well as a growing emphasis on school achievement [1,2]. In light of the prevailing notion of adolescence as a turbulent period of life, it is not surprising that much of the research on adolescent mental health has focused on negative aspects such as mental health problems and psychosomatic complaints [3]. 

During more recent years, however, research on adolescent mental health has increased its focus on positive aspects of health [3]. Life satisfaction is one important part of positive mental health, and considered as a vital component of the broader concept of subjective well-being [4,5]. Apart from being important for young people here and now, aspects of well-being in adolescence, such as the level of life satisfaction, may have long-term consequences well into adulthood. Indeed, longitudinal research on life satisfaction among adults has found that it predicts future health and even mortality [6]. Investigation into the correlates of life satisfaction at an early stage of life is therefore highly relevant. Previous research has reported associations between adolescent life satisfaction and various conditions at the individual [7,8,9], family [7,9,10,11,12], school [7,8,9,10,13,14,15,16], and national levels [12,17]. 

The present study focuses on the school as a central and modifiable social context for young people, and how conditions in this setting are linked to life satisfaction among adolescents in Sweden. Specifically, we are concerned with students’ psychosocial working conditions, namely perceived school demands, teacher support, and classmate support. School demands can be thought of as the perception of external and/or internal pressure to achieve goals established by teachers, parents or students themselves. Expectations put on students in the school environment are necessary for children’s learning and development [18]. However, students, even those who are high-performing, can experience school demands as unattainable [19], which might lead to stress [20]. Teachers are able to provide instrumental support to students, i.e., support that may aid academic achievements directly [21]. Additionally, teachers may also supply emotional support [22], which can affect perceptions of school belonging and have a positive impact on students’ self-esteem [23]. Support from classmates includes the feeling of acceptance in the school class and the belief that fellow students can offer assistance in a range of issues [24]. More broadly, peer relationships are crucial in adolescents’ lives, from the establishment of important friendships to the formation of one’s personal identity [24]. In general, social support from different sources have been shown to be associated with better well-being [25,26].

In research on adult populations, a common way of understanding the relationship between psychosocial conditions in the workplace and health is through the demand-control-support (DCS) model. This model postulates that high demands at work, coupled with low control and low support, cause the individual to experience stress and, by extension, poor health [27]. Studies on adolescent populations have applied the DCS model on psychosocial working conditions in school and several Swedish studies [28,29,30,31,32,33,34], as well as international ones [35], have consistently demonstrated that high school demands, as well as low teacher and classmate support, are correlated with worse health among students. 

The opportunities for schools to promote students’ well-being are plentiful, and such health interventions are important in the overall work towards enhancing mental health in today’s youth [36,37]. Poor functioning of the school has however been highlighted as one key factor behind the increase in psychosomatic complaints among Swedish adolescents during the past decades [38]. Still, less is known about the relationship between school-related conditions and aspects of positive health (for exceptions, see, e.g., [39,40]). Since the Swedish Education Act [41] requires the school health services to work with both health prevention and promotion, empirically grounded knowledge about the relationship between school-related conditions and life satisfaction among Swedish adolescents is especially pertinent. 

Using data drawn from a Swedish nationally representative sample, the aim of the present study was to examine differences in perceived psychosocial working conditions in school and life satisfaction by gender and by grade, and to scrutinize the association between psychosocial school working conditions and life satisfaction among boys and girls in Grades 5, 7, and 9. 

## 2. Methods

### 2.1. Data Material

Data from the Swedish Health Behavior in School-Aged Children (HBSC) study of 2017/18 were used. The HBSC study was performed among students in Grades 5, 7, and 9, which in the Swedish school system corresponds to ages ~11, 13, and 15. The data were collected using a two-step sampling design, first drawing a random sample of schools, and then randomly selecting one class in each school. The response rate was 47% for schools and 89% among students (*n* = 4264) [42]. The reason for the low response rate among schools was that not only students, but also schools, were anonymous due to legal reasons. Consequently, Statistics Sweden could not contact schools directly via telephone and remind them of the survey [42]. For the current study, we used information from 3614 students with valid information on all study variables (i.e., 85% of the total sample). 

### 2.2. Measures

Life satisfaction was measured using the 11-step Cantril ladder with the range 0–10. Previous studies have used the full scale [13,17], as well as dichotomous versions of the measure with cutoffs at values >5 [7,43] or >8 [5,7]. In the current study, we used the measure as a dichotomous variable (using both >5 and >8 as cut-offs), but we also performed analyzes with the variable as a continuous scale (presented in the Appendix A). 

School demands were constructed from three items: “How pressured do you feel by the schoolwork you have to do?”; “I find schoolwork difficult”; and “I have too much schoolwork”. Response categories for the first question were: “Not at all” = 1, “A little” = 2.33, “Quite a lot” = 3.67, and “A lot” = 5. For the latter two items, the response categories were: “Almost never” = 1, “Seldom” = 2, “Sometimes” = 3, “Often” = 4, and “Very often” = 5. Internal consistency was fairly high (Cronbach’s α = 0.75). The measure has been used previously [40]. 

Teacher support was based on three items: “I feel that my teachers accept me as I am”; “I feel that my teachers care about me as a person”; and “I feel a lot of trust in my teachers”. The response categories were “Strongly agree” = 5, “Agree” = 4, “Neither agree nor disagree” = 3, “Disagree” = 2, and “Strongly disagree” = 1. Internal consistency was high (Cronbach’s α = 0.87). The same set of items have been used in earlier studies [40,44].

Classmate support was measured by three items: “The students in my class(es) enjoy being together”; “Most of the students in my class(es) are kind and helpful”; and “Other students accept me as I am”. The response categories were “Strongly agree” = 5, “Agree” = 4, “Neither agree nor disagree” = 3, “Disagree” = 2, and “Strongly disagree” = 1. Internal consistency was high (Cronbach’s α = 0.80). The same set of items have been used in prior studies [40,44].

The scales measuring school demands, teacher support, and classmate support were calculated by using the mean score for participants who had responded to at least two out of three items within a scale. All scales ranged between 1–5, with higher values indicating higher levels of perceived school demands, teacher support, or classmate support. The pairwise correlation between school demands and teacher support was −0.37, between school demands and peer support −0.23, and between teacher support and peer support 0.42 (all correlations being statistically significant at *p* < 0.001). Additional analyzes using categorical versions of these measures are presented in the Appendix A. In these analyzes, the variables were coded into three categories of about equal size with the purpose of distinguishing between relatively high, intermediate, and low school demands, teacher support, and classmate support. 

Additionally, gender, grade, parents’ country of birth, and socioeconomic position measured by the family affluence scale (based on six items on the family’s number of cars, number of computers, number of bathrooms, ownership of a dishwasher, number of times travelled abroad on holiday during the last year, and whether or not the student has his or her own bedroom [45]) were included in the analyzes.

### 2.3. Statistical Analysis

Differences in proportions of students reporting high life satisfaction by gender and by grade were examined by cross-tabulations and chi-squared tests. For the main independent variables, i.e., school demands, teacher support, and classmate support, differences in mean values across gender and grade were examined using *t*-tests and ANOVA, respectively. The associations between psychosocial school working conditions and life satisfaction were evaluated by means of binary logistic regression analysis. Robust standard errors were estimated to account for the fact that students were clustered in classes. The number of classes was 213. We performed crude analyzes including one independent or control variable at a time, adjusting only for gender and grade, as well as adjusted models, mutually adjusting for all independent and control variables. Interaction terms between the independent variables and gender and grade, respectively, were evaluated using Wald tests. The tables present odds ratios (OR) with 95% confidence intervals (95% CI).

### 2.4. Ethical Considerations

Data from the Swedish HBSC study contain no information on personal identification and the questionnaires are filled in anonymously and voluntarily by the students. Since no sensitive data is collected, no formal approval from an ethical review board was required. Appropriate ethical consent was obtained in individual schools from the students who participated. Participating schools informed parents/guardians about the study beforehand and parents were asked to inform the school if they did not want their children to participate. 

## 3. Results

Descriptive statistics are presented in Table 1. In the study sample, the mean value of school demands was 2.87, of teacher support 4.10, of classmate support 3.93, and of family affluence 9.40. With regard to life satisfaction, 14.9% reported a score below 6 on the 11-step Cantril ladder, 54.2% a score between 6–8, and 30.8% a score above 8. The study sample included about equal shares of boys and girls, and comprised of 26.5% students in grade 5, 33.5% in grade 7, and 40.0 % in grade 9. About one fifth had two parents who were born outside Sweden. 

Differences in proportions of students reporting high life satisfaction as well as differences in mean values of school demands, teacher support, and classmate support by gender and by grade are presented in Table 2. Boys were more likely to report high life satisfaction than girls, regardless of cut-off point. The proportion of students reporting high life satisfaction was largest in Grade 5 and smallest in Grade 9, and this was true for boys and girls alike. Girls in Grade 9 were least likely to report high life satisfaction. With regards to school demands, girls reported higher levels than boys. There was also a gradual increase in levels of school demands by grade in the total sample and amongst both genders. Boys reported higher levels of teacher support than girls. There was a gradient pattern by grade with increasingly lower levels of teacher support in higher grades, among boys and girls alike. Finally, with regards to classmate support, boys reported higher levels than girls. For boys, there was no difference in classmate support by grade, whereas girls in Grades 7 and 9 reported lower levels of classmate support compared with those in Grade 5.

Table 3 displays results from the binary logistic regression analyzes with high life satisfaction (scores > 5 and >8, respectively) as dependent variables. In the crude analyzes, school demands were negatively associated with high life satisfaction, while both teacher support and classmate support were positively associated with high life satisfaction. These associations were attenuated but remained statistically significant in the fully adjusted analyzes. The crude analyzes also demonstrated that girls were less likely to report high life satisfaction than boys, that students in Grades 7 and 9 were less likely to report high life satisfaction than students in Grade 5, that respondents with both parents born outside of Sweden were more likely to report high life satisfaction than respondents with Swedish-born parents when scores >8 was used as cut-off, and that higher family affluence correlated with a higher likelihood of reporting high life satisfaction. In the adjusted analyzes, gender and family affluence were still associated with high life satisfaction, having both parents born outside of Sweden was associated with high life satisfaction regardless of cut-off point, and only students in Grade 9 when high life satisfaction was defined as >8 were less likely to report high life satisfaction. 

Next, we included interactions between the main independent variables, i.e., school demands, teacher support, and classmate support, and gender and grade, one at a time (not displayed in table). Using Wald tests, two statistically significant interactions were found when >5 was used as cut-off point for life satisfaction, namely between grade and school demands (Grade 9 × school demands: OR = 1.63, *p* = 0.012), as well as between grade and classmate support (Grade 7 × classmate support: OR = 0.59, *p* = 0.010; Grade 9 × classmate support: OR = 0.50, *p* < 0.001), indicating that the associations were stronger for younger students. 

Subsequently, analyzes stratified by gender and grade, respectively, were performed, with results presented in Table 4. These revealed that school demands, teacher support, and classmate support were associated with life satisfaction among both boys and girls and among students in all grades. As indicated by the statistically significant interaction terms, the associations that life satisfaction (using the cut-off >5) shared with school demands and classmate support were stronger among students in Grade 5. 

As an additional check, we also conducted analyzes where the three independent variables of interest were divided into three groups of about equal size, distinguishing between relatively “low”, “intermediate”, and “high” levels of school demands, teacher support, and classmate support. The results from these analyzes are presented in Table A1 in the Appendix A. Furthermore, we performed linear regression analyzes with life satisfaction as a continuous measure, with results presented in Table A2 in the Appendix A. Even though the linear model indicated problems with heteroskedasticity, results from these additional analyzes reflect the same pattern as presented above, with clear and consistent links between our measures of psychosocial school working conditions and life satisfaction among boys and girls and among students in all grades.

## 4. Discussion

The current study examined the associations between life satisfaction and psychosocial working conditions in terms of school demands, teacher support, and classmate support, among students in Sweden. Given that Swedish schools are required by law to work with both health prevention and health promotion [41], knowledge about modifiable school-related correlates of life satisfaction among Swedish adolescents seems vital. 

The results showed that school demands were inversely associated with high life satisfaction, while teacher and classmate support were positively associated with high life satisfaction. These results corroborate previous studies from other countries that have reported associations between life satisfaction and school-related factors such as schoolwork pressure, school stress, and strain [13,15,16], teacher support [9], and student relations [16]. We also demonstrated that these associations were found for both boys and girls and among students in all studied grades. Consistent with prior research [5], the study also showed that boys reported higher life satisfaction than girls, and that life satisfaction was highest among students in Grade 5 and lowest among those in Grade 9. Furthermore, whereas girls reported higher school demands than boys, higher levels of teacher and classmate support were reported by boys. Students in lower grades reported lower school demands but higher levels of teacher and classmate support compared with students in higher grades. 

There might be several mechanisms at work that could explain the associations between our working conditions indicators and life satisfaction. One interpretation is that perceiving the demands in school as high and/or the social support as low may be an indication of students’ self-esteem or perceived competence in relation to academic or social endeavors [9,14,16], which, in turn, may impact life satisfaction [7]. The experience of high levels of stress may be another interpretation of our findings, as high levels of school demands can be an indication that the student experiences stress to a high degree, which might impair life satisfaction [15]. Having supportive relationships may however work as a buffer against stress, e.g., through the provision of instrumental or emotional support [25], and thereby promote life satisfaction. Support from both teachers and classmates can also function as protection against other types of factors that could impact life satisfaction negatively, such as health risk behaviors [16] or school dissatisfaction [8]. Above the protective quality of social support, positive relationships with others at school can also directly promote life satisfaction through feelings of belonging and companionship [25], and the construction of a social identity [46]. 

Taken together, the findings indicate that schools have the potential to promote students’ positive health. The results highlight the importance of schools helping their students to handle the demands placed on them, shaping an environment that creates opportunities for teachers to provide adequate support for the students, and promoting a benign and supportive social climate among students [40]. Health intervention programs at schools can target a range of indicators in the realm of mental health, including positive indicators that contribute to a healthy development of adolescents, and there is evidence that these interventions benefit children of all ages [36,47,48]. It is clear that the associations examined in the current study were valid for boys and girls in all grades, indicating that interventions may be relevant and impactful for all students.

While the positive association between social support and life satisfaction is expected, and the notion of promoting a supportive environment at school is uncontroversial, the potential issue concerning school demands may warrant more discussion. To reach learning and developmental goals, schools have to place expectations on students’ academic achievements, and these demands may have a positive impact on students’ future health status through cognitive development and opportunities for further education. However, the potential negative consequences of high school demands should be considered and studied further.

A major strength of the study is the fact that it was based on large-scale data from a nationally representative sample of Swedish students in Grades 5, 7, and 9. Yet, there are also shortcomings. One limitation concerns the cross-sectional nature of the data, which refrains us from drawing conclusions about causality with support in the data. It is likely that psychosocial working conditions in school affect adolescents’ life satisfaction, but it is also possible that students with low life satisfaction tend to perceive school demands as higher and support from teachers and classmates as lower. Future studies should disentangle the links between school-related conditions and aspects of positive health among students using data from different points in time. Moreover, there may be other relevant factors influencing both students’ perception of their psychosocial working conditions in school and their life satisfaction that we did not include in the analyzes, such as personality traits, conditions in the family, and regional factors. Future studies should address this. There is also room for more qualitatively orientated and mixed-methods approaches, which may elucidate how psychosocial working conditions at school are related to life satisfaction and other aspects of positive health among students. Furthermore, the data suffered from significant non-response at the school level, and since we cannot rule out the risk of systematic bias in the non-response, generalizations to all Swedish students in the studied grades need to be made with caution.

## 5. Conclusions

Using large-scale data collected amongst Swedish students in Grades 5, 7, and 9, the current study showed clear and consistent associations between psychosocial working conditions in terms of school demands, teacher support, and peer support, and students’ life satisfaction. Thus, the findings indicate that schools have the potential to promote students’ positive health. More specifically, the results suggest that schools may enhance students’ positive health by helping them to handle school demands, by enabling teachers to provide relevant support, and by working towards a favorable social environment among the students. Policies and interventions targeted at adolescent mental health may benefit from not only focusing on negative aspects of health, such as health complaints and problems, but also on promoting positive aspects of health and the state of happiness among adolescents. Such pursuits could achieve good results by focusing on the psychosocial working environment of schools.

## Figures and Tables

**Table 1 ijerph-18-05337-t001:** Descriptive statistics (*n* = 3614).

Variables	Mean	S.D.	Min.	Max.
School demands	2.87	0.93	1	5
Teacher support	4.10	0.86	1	5
Classmate support	3.93	0.77	1	5
Family affluence scale	9.40	1.94	1	13
	*n*	%		
Life satisfaction				
Scores 0–5	541	14.9		
Scores 6–8	1960	54.2		
Scores 9–10	1113	30.8		
Gender				
Boys	1768	48.9		
Girls	1846	51.1		
Grade				
5	959	26.5		
7	1210	33.5		
9	1445	40.0		
Parents’ country of birth				
Sweden	2886	79.9		
Other	728	20.1		

**Table 2 ijerph-18-05337-t002:** Life satisfaction and psychosocial school conditions by gender and grade (*n* = 3614).

Variables	Life Satisfaction (Scores > 5)		Life Satisfaction (Scores > 8)		School Demands		Teacher Support		Classmate Support	
	%	X^2^	%	X^2^	Mean		Mean		Mean	
Gender										
Boys	89.3		36.1		2.70		4.17		4.01	
Girls	81.0	48.49 ***	25.7	45.43 ***	3.03	*p* < 0.001	4.03	*p* < 0.001	3.85	*p* < 0.001
Grade										
5	90.3		43.1		2.27		4.43		3.99	
7	85.5		32.9		2.81		4.08		3.92	
9	81.1	38.66 ***	20.9	136.64 ***	3.30	*p* < 0.001	3.90	*p* < 0.001	3.89	*p* < 0.01
Gender and grade										
Boys Grade 5	92.2		46.3		2.23		4.40		4.00	
Boys Grade 7	90.4		39.5		2.63		4.16		4.02	
Boys Grade 9	86.1	12.19 **	25.6	58.14 ***	3.10	*p* < 0.001	4.02	*p* < 0.001	4.01	n.s.
Girls Grade 5	88.3		39.6		2.32		4.46		3.99	
Girls Grade 7	80.9		26.5		2.99		4.00		3.83	
Girls Grade 9	76.7	25.01 ***	16.8	78.36 ***	3.48	*p* < 0.001	3.79	*p* < 0.001	3.79	*p* < 0.001

*** *p* < 0.001 ** *p* < 0.01.

**Table 3 ijerph-18-05337-t003:** Odds ratios and 95% confidence intervals from binary logistic regressions of life satisfaction (*n* = 3614).

Variables	Life Satisfaction (Scores > 5)				Life Satisfaction (Scores > 8)			
	Crude ^a^		Adjusted ^b^		Crude ^a^		Adjusted ^b^	
	OR	95% CI	OR	95% CI	OR	95% CI	OR	95% CI
School demands	0.48 ***	0.42–0.54	0.61 ***	0.53–0.69	0.57 ***	0.52–0.64	0.70 ***	0.62–0.78
Teacher support	2.13 ***	1.92–2.37	1.60 ***	1.42–1.80	1.97 ***	1.74–2.24	1.45 ***	1.27–1.65
Classmate support	2.42 ***	2.10–2.79	1.81 ***	1.54–2.12	2.33 ***	2.06–2.65	1.87 ***	1.64–2.14
Gender								
Boys (ref.)	1.00	-	1.00	-	1.00	-	1.00	-
Girls	0.52 ***	0.42–0.64	0.67 ***	0.54–0.83	0.62 ***	0.53–0.73	0.74 ***	0.62–0.87
Grade								
5 (ref.)	1.00	-	1.00	-	1.00	-	1.00	-
7	0.64 **	0.47–0.88	1.00	0.72–1.38	0.65 ***	0.53–0.81	0.87	0.69–1.08
9	0.47 ***	0.35–0.64	1.02	0.72–1.44	0.35 ***	0.28–0.45	0.55 ***	0.44–0.68
Parental country of birth								
Sweden (ref.)	1.00	-	1.00	-	1.00	-	1.00	-
Other	0.95	0.75–1.21	1.40 *	1.08–1.82	1.58 ***	1.30–1.91	1.85 ***	1.50–2.28
Family Affluence Scale	1.25 ***	1.19–1.31	1.27 ***	1.20–1.34	1.06 **	1.02–1.11	1.09 ***	1.04–1.14
Pseudo R^2^	-		0.18		-		0.13	

*** *p* < 0.001 ** *p* < 0.01 * *p* < 0.05, ^a^ adjusted only for gender and grade. ^b^ Mutually adjusted for all independent variables.

**Table 4 ijerph-18-05337-t004:** Odds ratios and 95% confidence intervals from binary logistic regressions of life satisfaction, stratified by gender and age, respectively.

Variables	Life Satisfaction (Scores > 5)				Life Satisfaction (Scores > 8)			
	Crude ^a^		Adjusted ^b^		Crude ^a^		Adjusted ^b^	
	OR	95% CI	OR	95% CI	OR	95% CI	OR	95% CI
Boys (*n =* 1768)								
School demands	0.52 ***	0.43–0.62	0.66 ***	0.54–0.82	0.61 ***	0.52–0.71	0.72 ***	0.62–0.84
Teacher support	2.03 ***	1.74–2.37	1.60 ***	1.33–1.93	1.86 ***	1.58–2.18	1.45 ***	1.23–1.72
Classmate support	2.33 ***	1.89–2.86	1.72 ***	1.36–2.17	2.09 ***	1.76–2.47	1.69 ***	1.42–2.01
Girls (*n =* 1846)								
School demands	0.45 ***	0.38–0.54	0.57 ***	0.47–0.68	0.54 ***	0.46–0.64	0.67 ***	0.57–0.78
Teacher support	2.22 ***	1.93–2.54	1.60 ***	1.36–1.88	2.11 ***	1.75–2.55	1.43 ***	1.18–1.73
Classmate support	2.49 ***	2.09–2.96	1.89 ***	1.56–2.30	2.64 ***	2.19–3.19	2.13 ***	1.75–2.58
Grade 5 (*n =* 959)								
School demands	0.35 ***	0.25–0.49	0.45 ***	0.31–0.66	0.57 ***	0.47–0.71	0.73 **	0.59–0.90
Teacher support	2.61 ***	2.09–3.28	1.67 **	1.24–2.25	2.11 ***	1.72–2.59	1.44 **	1.14–1.81
Classmate support	3.62***	2.63–4.97	2.90 ***	2.11–4.00	2.42 ***	1.95–3.00	1.97 ***	1.59–2.44
Grade 7 (*n =* 1210)								
School demands	0.47 ***	0.38–0.58	0.60 ***	0.48–0.76	0.56 ***	0.47–0.67	0.64 ***	0.54–0.76
Teacher support	2.22 ***	1.78–2.77	1.67 ***	1.30–2.16	1.77 ***	1.41–2.23	1.38 **	1.10–1.74
Classmate support	2.36 ***	1.88–2.96	1.74 ***	1.34–2.26	1.93 ***	1.59–2.35	1.59 ***	1.29–1.97
Grade 9 (*n =* 1445)								
School demands	0.55 ***	0.47–0.64	0.68 ***	0.58–0.81	0.59 ***	0.49–0.72	0.72 **	0.59–0.88
Teacher support	1.95 ***	1.72–2.22	1.57 ***	1.37–1.80	2.10 ***	1.71–2.58	1.52 ***	1.22–1.88
Classmate support	2.16 ***	1.76–2.65	1.56 ***	1.26–1.95	2.78 ***	2.22–3.49	2.15 ***	1.68–2.76

*** *p* < 0.001 ** *p* < 0.01, ^a^ adjusted only for grade. ^b^ Mutually adjusted for all independent variables.

## Data Availability

The Swedish HBSC data of 2017/18 can be applied for at the Public Health Agency of Sweden. Data from previous waves in Sweden and in other participating countries is available at: https://www.uib.no/en/hbscdata (accessed on 17 May 2021).

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
