# Peer review of "Psychosocial Working Conditions in School and Life Satisfaction among Adolescents in Sweden: A Cross-Sectional Study"

_ijerph, 2021, doi:10.3390/ijerph18105337_

Round 1
Reviewer 1 Report
Some methodological errors have been made in the paper that affect the regression models presented.
Regarding the regression analysis performed and shown in table 3, it should be pointed out that the crude model is not valid when there are other independent variables affecting the response variable. In other words, only the so-called adjusted model should be presented.
As for the analysis stratified by gender, the results of which are presented in table 4, it makes sense if there is interaction between gender and the other independent variables in the model. This does not seem to be the case, but it is only possible to know this by introducing such interactions in the initial model. From such an analysis it is possible to determine which interactions are significant and how these variables are affected by gender.
The same can be said about the stratified analysis by grade, it only provides interesting additional information if there are significant interactions between the grade variable and the other independent variables.
On the other hand, the models presented in the appendix (appendix tables 1-4) present the same problems as discussed above, so a review of these models is necessary.
Moreover, it should be noted that the response variable "life satisfaction" is measured on an ordinal scale, so that an ordinal or multinomial regression model might be more appropriate than the linear model analysed. In any case, it would be interesting to carry out a diagnosis of the model to check the suitability of this linear model.
Finally, the coefficient of determination obtained may indicate that there are other factors influencing the response variable that have not been taken into account in the model. A reflection on this aspect should be included in the Discussion section.
Reviewer 2 Report
This brief has been addressed to examine the relationship between some school conditions and the life satisfaction reported by young students through a questionary.
It is a well-written work, the results are detailed and it is centered on a necessary topic: the well-being in terms of life satisfaction of adolescents and the role of schools in its promotion. Nevertheless, possible generalisations, causations or determinist relations should be carefully revised.
Some specific comments/recommendations are posed in order to improve the suitability of the proposal:
- Abstract: to initiate this section with some hints about the state of the art could be clarifier.
- Introduction: it would be positive to include in this framework the definition and state of the art of the “school working conditions” that are going to be analyzed in the report (i.e. school demands, teacher support, classmate support). As it is a current emerging debate, related to high school demands, it would be of interest to specify, how scientific evidence is more and more stressing the potentialities of high expectations and effort for the students’ cognitive development in relation to health.
-
Methods: “Measures” section should be typed as a subsection of Methods. It would be of interest to include an “ethical considerations” section detailing how it was applied along the whole process. Informed consent forms of students’ participation were provided by families? Please, reference the scale used in the study.
-
Discussion: it should be displayed in a separate line (now is attached to the last Word in the results section). More evidence about the school as a well-being promoter, on the one hand, and on the fact of quality interactions as predictor of life satisfaction, on the other hand, could be provided in this section. It is important to underline the correlation between factors, avoiding casual explanations which would be unreal as they cannot be derived from the study. Maybe a mixed methods approach, considering the voices of the subjects through qualitative techniques could be a future line of research for shedding more light to these correlations.
-
Conclusions: again, in this point, it should be avoided to present results as predictive, instead of correlations. More detailed presentation of the conclusions would be of interest.
-
References: Please, check that all the references are presented according to the Reference List and Citations Style Guide for MDPI Journals, including their DOI.
Round 2
Reviewer 1 Report
First of all, the authors' effort to follow the indications of the first review should be appreciated. However, some methodological errors are still made that cannot be overlooked.
As the authors point out, the main change in the manuscript is that they now perform two logistic regressions using two different cut-off points. The reason given is that the estimates are more interpretable with dichotomous dependent variables. This is a very subjective opinion. Ordinal or multinomial regression are well-known techniques that could have been used to avoid doing two logistic regressions. Nevertheless, the results obtained are in line with those presented for the linear regression model (now included in the appendix). In my previous review I indicated the possibility of making a diagnosis of such a model, which could indicate whether it was adequate, especially considering that the Cantril scale takes values from 0 to 10. Although the authors indicate that they have made such a diagnosis, they do not present any results on it. In summary, although the most appropriate analysis would be to consider an ordinal regression or the linear model if the diagnosis of the model is satisfactory, the use of logistic regression can be accepted given that the results found are consistent with those of the linear model.
On the other hand, point 1 indicated that the crude model was not valid, so it should be eliminated from the paper. It is a methodological error not to take into account all those variables that significantly affect the dependent variable. For example, one of the consequences of this is the possibility of spurious correlation. It is true that this is not the case and that the results of the crude model are very similar to those of the adjusted model, but rigorous work should not include analyses that are not appropriate.
It was also pointed out in points 2 and 3 that the stratified analysis by gender and grade does not make sense if there are no interactions. The authors indicate that there are two interactions that are significant. In that case they should be included in the initial model to appreciate their effect. In my opinion the stratified analysis does not provide much information if the model includes the interactions, but they could be kept to complement the analysis.
Finally, the possibility of other factors influencing the response variable, as the authors themselves acknowledge, may affect the results obtained with the estimated model. The problem is that the relationships found between the variables considered and life satisfaction may disappear in the presence of these other omitted variables. Unfortunately, this is a problem that always surrounds these experimental studies and may invalidate the conclusions found or at least be very cautious in generalising the results obtained.
Reviewer 2 Report
Dear authors:
Thank you for considering the suggestions made in the review.
The revised version is being appropiated for publication. Nevertheless, it would be desirable to include the following clues that were recommended in the first review and have not been included yet in the revised version:
- Clear definitions of the “school working conditions” analyzed in the report. In order to have a greater comprehension of the study, there should be specified what the researchers understand as "school demands", "teacher support", and "classmate support", according to previous studies.
- When reporting science, it is always important to clarify all the evidences known until the present moment regarding a specific topic. In the case of this study, it is providing knowledge to the existing literature about the current emerging debate, related to high school demands, while scientific evidence is more and more stressing the potentialities of high expectations and effort for the students’ cognitive development in relation to health. It is very important to pose this question in the paper, either in the introduction, or in the discussion section.
It is encouraged to contemplate these suggestions in order to let a thorough understanding of the study for readers.
